# Analysis of factors influencing levee safety using the DEMATEL method

**Dorota Mirosław-Świątek**[1]*, **Paweł Popielski**[2], **Piotr Śliwiński**[3], **Tomasz Cwalina**[4], **Zdzisław Skutnik**[5]

**1** Department of Hydrology, Meteorology and Water Management, Institute of Environmental Engineering, Warsaw University of Life Sciences—SGGW, Warsaw, Poland, **2** Faculty of Building Services, Hydro and Environmental Engineering, Warsaw University of Technology, Warsaw, Poland, **3** Office of Geotechnics, Filtration and Concrete Quality of Hydroengineering Structures, Dam Technical Supervision Centre, Institute of Meteorology and Water Management IMGW- PIB, Warsaw, Poland, **4** 18-400, Stara Łomża, Poland, **5** Institute of Civil Engineering, Warsaw University of Life Sciences-SGGW, Warsaw, Poland

* dorota_miroslaw_swiatek@sggw.edu.pl

**Data Availability Statement:** All relevant data are within the paper and its Supporting information files.

**Funding:** The author(s) received no specific funding for this work.

## Abstract

River embankments are the basic and the oldest measures of protecting areas potentially subjected to flooding, and at the same time pose a serious threat to their environment in the event of damage or failure. The technical condition of the levees and its regular evaluation is a key element of their safety. A general assessment of the technical condition of a levee is the result of many interacting factors and parameters that depend on each other to a varying degree. Therefore, it is necessary to consider the cause-and-effect links between the inter-relationships of numerous parameters and sensors of significant impact. In this article the decision-making trial and evaluation laboratory (DEMATEL) method was applied to develop a cause-and-effect model for factors impacting the condition and safety of levees. Effective factors impacting the technical condition of a levee were identified; relationships between these factors were determined; a cause-and-effect model was developed based on identified factors; factors were categorized based on the dependence scale and influential indicators of each factors used in the DEMATEL method. The obtained results demonstrate that three following factors: hydrological factor, type and condition of soils in levee body and condition of levee areas (inter-levee and landside) play the most important role for levee safety. The results of this study can support traditional assessments of hydrotechnical structure or assist entities managing levees.

## Introduction

Levees are the basic and the oldest measures of protecting flood-endangered areas. Like most hydrotechnical structures, levees pose a serious threat to its environment in the event of damage or failure [1–4]. Levees are structures periodically damming up water, which means that they are subjected to their design loads only in the event of a floodflow. Their technical condition is a crucial element affecting their safety. Hence, regular evaluation of their technical condition and safety level is extremely important, all the more because it is conducted in periods

**Competing interests:** The authors have declared that no competing interests exist.

out of floodflows. Most countries develop detailed guidelines regarding the maintenance, periodic and extraordinary inspection of levee condition and work associated with their protection in the event of flood waters [5–8]. In various socio-economic scenarios, global warming will most probably lead to increased flood-related threat in Europe, hence, both levees and other structures will be breached more often [9].

An evaluation of the technical condition and safety of levees is preceded by a significant number of measurements, studies, analyses and observations. It is necessary to perform geodetic surveying (including laser scanning), geotechnical studies with sampling for laboratory tests, as well as geophysical measurements and tests (microgravimetric, seismic or electroporous tomography) [6, 10, 11]. A visual assessment of the condition of embankment structures, including the measurement of their material properties is required. Consecutively, based on obtained data, e.g., soil parameters, calculations are made that cover filtration and the stability of the levee and its subsoil, taking into account river hydrology in defined sections. All of the above actions lead to determining factors that can influence the technical condition of an assessed levee section. Geodetic surveying enable determining the current levee dimensions and geometry changes (relative to archival or design measurements), and the geotechnical tests allow to assess the condition of the soil that make up the levee body or the soils of the bed, inter-levee or land side, and to determine their parameters. Whereas assessing the condition of the levee structures is important in terms of their impact on the general technical condition and the possible occurrence of phenomena on concrete element interfaces, e.g., levee culvert with the soil structure of the levee. Observations and visual assessment make it possible to identify animal activity and the significance of their impact on the structure. Summing up, it can be concluded that a general assessment of the technical condition of a levee is the resultant of many interacting factors and parameters that depend on each other to a varying degree.

When analysing levee condition, it is necessary to consider the cause-and-effect links between the interrelationships of numerous parameters and sensors of significant impact. Although the very factors impacting the technical condition and their analysis when assessing the technical condition and safety of levees are widely discussed in sources literature [5, 6, 8, 12, 13], their inter-relationships have never been previously described in the form of a cause-and-effect model. Such knowledge can support traditional assessments of hydrotechnical structure or assist entities managing levees. Owing to these models, engineers or decision-makers can focus on more important elements or make the right decisions. They can also be used in education and information campaigns covering flood protection. The decision-making trial and evaluation laboratory (DEMATEL) method has been successfully used to address such issues associated with developing a cause-and-effect model [14–16]. It enables to determine the influential power and the relationship scale for each factor and to visualise such a structure through matrices or digraphs, based on relationships between effective factors. This method not only transforms the interdependence relationships into a cause-and-effect group using a matrix, but also finds the critical factors of a complex system using an influence relation diagram. It has been applied for solving the problems of complex systems in various fields, such as business management, product and service shaping, transport, power, medicine, finances, banking, education, IT systems, environmental engineering and construction [17]. For example, it has been used in respect of the risk analysis of technical facilities in the work [18], where the authors identified the causes of a concrete structure failure. It has also found applications as an effective tool for the risk analysis of a tunnel engineering structure [19].

This article thoroughly describes the cause-and-effect relationships between factors influencing the technical condition and safety of levees, through achieving the following objectives: i) identifying effective factors influencing levee technical condition; ii) determining relationships between these factors through the DEMATEL method; iii) developing a cause-and-effect

model based on the identified factors; iv) categorizing factors based on the influential indicators and the relationship scale for each factor used in the DEMATEL method.

## Materials and methods

### DEMATEL method

As mentioned before, DEMATEL is a comprehensive method that allows to analyse complex system interrelationships. This method transforms interrelationships between factors into an understandable structural model of a system and divides them into a group of causes and a group of effects. A finished cause-and-effect model can be presented in the form of a diagram. Classic DEMATEL method stages are shown (Fig 1) [20, 21].

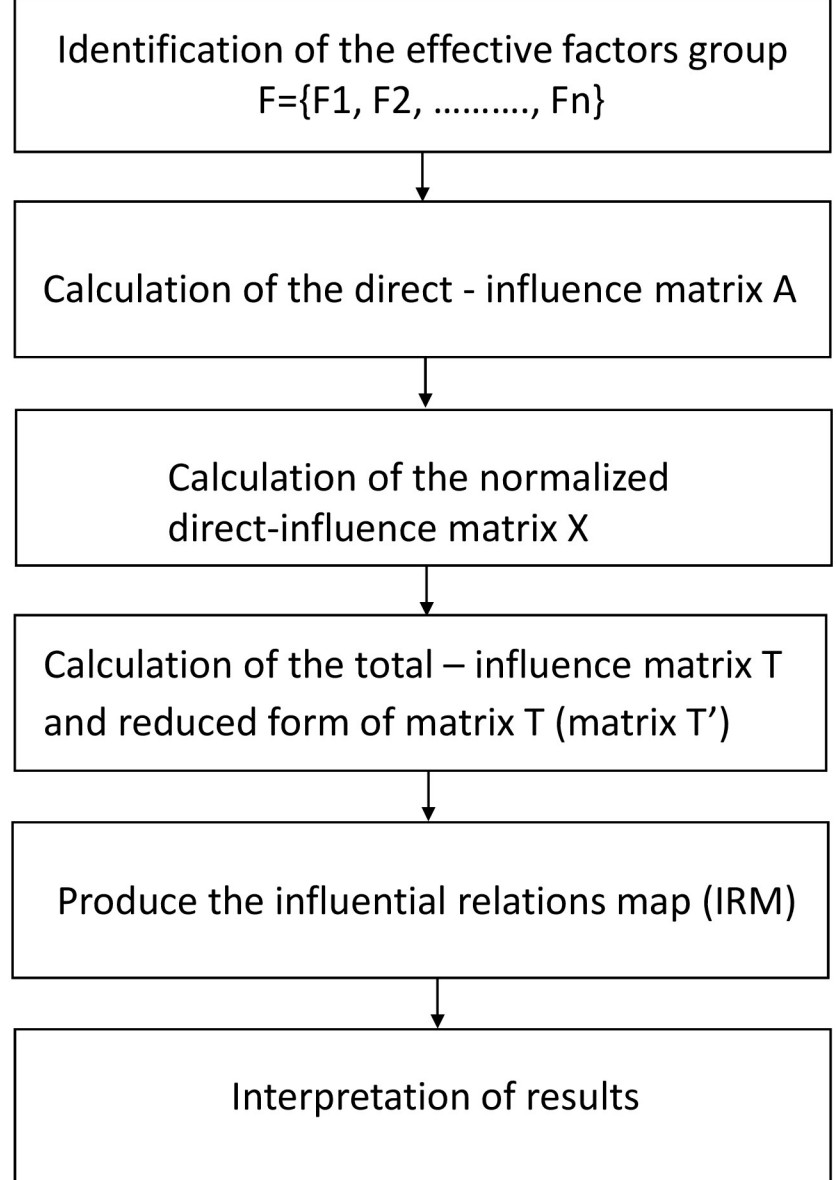

**Fig 1. Flow chart for calculation steps in the DEMATEL method.**

The procedures for implementing DEMATEL are as follows:

**Step 1.** A group of effective factors F = {F1, F2, . . ..., Fn}, with significant impact on functionality should be identified based on literature review and reception of expert opinions.

**Step 2.** Experts formulate the direct-influence matrix $A = [a_{ij}]_{n \times n}$ by indicating the direct influence that the factor $F_i$ has on $F_j$, using an integer scale (0 –N) of: no influence (0); low influence (1); medium influence (2); high influence (3). The $A$ matrix is determined under an assumption that each factor can directly influence other elements but cannot impact itself, which means that all principal elements of the $A$ matrix are equal to zero. Values of the direct—influence matrix elements can be jointly agreed by experts [18] or calculated as a mean value based on individual direct-influence matrix, provided independently by each expert [17, 22].

**Step 3.** Using the direct-influence matrix, the normalized direct-influence matrix $X = [x_{ij}]_{n \times n}$ is calculated by

$$X = \frac{A}{s} \tag{1}$$

$$s = max \left( max_{1 \leq i \leq n} \sum\nolimits_{j=1}^{n} a_{ij,} \ max_{1 \leq j \leq n} \sum\nolimits_{i=1}^{n} a_{ij,} \right) \tag{2}$$

**Step 4.** Based on matrix $X$, the total-influence matrix $T = [t_{ij}]_{n \times n}$ is calculated by summing the direct effects and all of the indirect effects by

$$T = X(I - X)^{-1} \tag{3}$$

where: $I$–identity matrix;
The total influence matrix $T$ contains information on the intensity of the total mutual influence of the factors.

**Step 5.** The influential relation map (IRM) is created by mapping a set of two indexes (R+C, R-C) based on two vectors $R$ and $C$, representing the sum of rows and sum of the columns from the total-influence matrix. These vectors are defined by the following terms:

$$R = [r_i]_{n \times 1} = \left[ \sum\nolimits_{j=1}^{n} t_{ij} \right]_{n \times 1} \tag{4}$$

$$C = \left[ c_j \right]_{1 \times n} = \left[ \sum\nolimits_{i=1}^{n} t_{ij} \right]_{1 \times n} \tag{5}$$

The $r_i$ (influential power) indicator is a sum of the $i$-th row of matrix $T$ and describes the sum of direct and indirect influences (effects) of factor $i$ on other factors. Likewise, $c_j$ (dependency) is the sum of the $i$-th column in matrix $T$ and presents the sum of indirect and direct effects of matrix $T$, which factor $F_j$ receives from other factors. Therefore, it is the value of the total, direct or indirect, effect that is exerted by factor on the analysed factor. The *Prominence* (R+C) and *Relation* (R-C) indicators determined for each $F_i$ based on $r_i$ and $c_i$ are used to unambiguously assign factors to a group of causes or effects. *Prominence* describes the strength of influence given and received by a given factor. The value $r_i+c_i$ means a role that a given factor plays within the system in the process of determining factor

nature. The *Relation* shows the net effect that a given factor brings into the system and is the basis for ranking factors. If $r_i$-$c_i$ is positive, then $F_i$ belongs to a group of causes (impact the system). If $r_i$-$c_i$ is negative, then $F_i$ is the effect of net impact of other system elements and is classified in the group of effects. IRM enables the visualization of complex causal relationships among factors. Usually, the factors in complicated system are grouped, depending on the value of R+C and R-C, into four quadrants according to their locations in the IRM diagram [22–24]. Due to their location in a specific quadrant, factors are classified as: most important, important, independent, indirectly.

**Step 6.** The net influence matrix–$Net = [Net_{ij}]_{n \times n}$ [25, 26] is used to evaluate the influential strength of one factor over another. *NeT* elements describe the net influence value of a given factor. The *NeT* matrix is calculated from the total influence matrix $T$, according to the following formula:

$$Net_{ij} = t_{ij} - t_{ji} \qquad (6)$$

The total net influence matrix can be used to develop an acyclic and asymmetric directed graph called the total net influence map, which shows impact relationships between individual factors.

**Step 7.** The resulting total influence structure is often of very complex nature since it represents all system element interrelationships. Reduction is used for simplification. It enables filtering out irrelevant relationships through eliminating relationships from matrix $T$ that exhibit value lower than the adopted positive total influence threshold θ. The resulting structure is called a reduced total influence structure and is expressed in the form of matrix $T' = \left[ t_{ij}^{\cdot} \right]_{n \times n}$, the elements of which are calculated as follows:

$$t_{ij}^{\cdot} = \begin{cases} t_{ij} & t_{ij} > \theta \\ 0 & t_{ij} \leq \theta \end{cases} \qquad (7)$$

In source literature, the threshold value θ is usually determined by expert methods [17] or the mean of all elements in the matrix $T$ [27].

## Basic factors influencing levee technical condition

The methods for assessing the technical condition and safety of levees that have been used for many years are based on broadly understood monitoring, which is the collection and analysis of qualitative and quantitative information and measured data or the observation of phenomena in an organized manner [28]. Regardless of the complexity of a measuring and inspection system, the results of such analyses lead in each case to assessing various physical quantities, which are factors with greatest impact on the technical condition of structures. Given the above, numerous recommendations and guidelines have been drawn up [2, 5, 7, 10, 29] used as a base to take measurements and conduct tests, followed by an assessment of the technical condition. After analysing the factors referred to in source literature most frequently, with greatest impact on the safety of levees, eight following effective factors were distinguished to be used for further analyses:

**Levee geometry** interpreted as a set of all levee dimension, defining its spatial shape, i.e., base width, height, top height, slope inclination and length, plan layout. These dimensions result from decisions made by a designer, based on available information on the type of soil

intended for its construction, and the type and condition of the ground. A change in the cross-section shape of a levee is often one of the first symptoms of its degradation [29].

**Type and condition of soils in levee body** described by a set of all data specifying the material making up a levee, i.e., soil types divided into strata, taking into account the soil condition and characteristic geotechnical parameters, i.e., density, compaction degree, cohesion, internal friction angle, filtration coefficient, plasticity degree, compressibility modulus, etc. The type and condition of soils in levee body is a result of many various factors. Most levees were built many years ago. The question on the most frequent kind of soil used to build levees most often comes with an answer "local", hence, the type and condition of soils in the bed can significantly impact their condition in the levee body. This, in turn, may result in the condition of the levee top and slopes being very dependent on the soils built into the levee body, their arrangement and compaction degree.

**Type, arrangement and condition of soils in levee bed** interpreted as a set of any data identifying the material making up a levee bed, i.e., types of soil divided into strata, taking into account their sequence from the levee foundation level, characteristic geotechnical parameters of the soils, i.e., density, compaction degree, cohesion, internal friction angle, filtration coefficient, plasticity degree, compressibility modulus, etc. Already at the design stage, the bed has a decisive influence on levee body shape (height, waterside and landside slope inclinations, top width). It also largely determines the arrangement of individual soil types within the cross-section, and impacts their condition within the levee body. The geological and hydrogeological conditions also have a significant influence on the condition of levee areas. It is very difficult to maintain the required levee top elevation and appropriate levee slope shape in the case of diversified bed structure.

**Condition of levee areas (inter-levee and landside)** determined by its topography, soil stratum arrangement in the bed—divided into strata, taking into account their sequence from the inter-levee surface, historical excavation locations of levee building material, and existing farm wells that disturb the terrain surface structure. The low and high vegetation development level, which impact potential damming or flow concentration in the event of a floodflow and motion resistance, are also of significance within the inter-levee area. Numerous trees and shrubs growing in the inter-levee area lead to water damming, the formation of water vortices and concentration of flow between the tree line and levee slope, resulting in washouts and damage to the waterside slope. Local damming can lead to water overflowing through the levee top. Trees growing near a levee or even on it facilitate the formation of filtration routes along their root system. Agricultural cultivation of inter-levee areas also has an adverse effect on the condition of levees. This leads to loosening of the top soil layer (usually poorly permeable), which facilitates the filtration through levee bed. These processes are also favoured by the presence of potholes, ponds and farm wells on the landside, and the inter-levee area.

**Condition of levee structures** understood as information on the technical condition of levee structures, such as culverts, sluices, siphons, etc. This information is related to the patency, settlement, structural continuity (concrete cracks) and closure efficiency, as well as the possibility of privileged filtration routes, especially at the concrete and soil interface. Quite often, the technical condition of levee structures, such as culverts, sluices, siphons, etc. is associated with the condition of the levee bed and body—it results from the load bearing capacity and deformability of the structure and bed soil. Maintenance of levee structures and levees should be conducted also in calm periods that do not experience floodflows.

**Activity of animals**, both wild and farmed–all information on the species of animals residing within the levee, their activity, impact on the structure, e.g., creating paths, burrows, tree felling, passages through the levee, etc. Animal activity is a rather significant destructive factor. Holes, nests and channels made by burrowing animals result in a serious risk to levee stability.

Foxes and beavers that dig extensive dens and corridors have also been contributing to this damage over the recent years. Besides burrowing animals, also farm animals impact the condition of levees. They are driven to inter-levee areas through the levee in prohibited places, which leads to slope turf damage. In extreme situations, cattle are grazed on the levees themselves.

**Condition of levee top and slopes** includes all available information on the quality of maintenance work, potential or actual filtration deformations [30], geometry changes (top settlements, displacements, slope soil slides and falls), damaged turf or protecting elements, etc. One of the most common causes for levee breaking is water overflowing the levee top, which leads to slope washout, followed by levee destruction. The main reason behind overflowing water is the low levee top level or the formation of local water damming due to inter-levee clogging [31]. The lack of even the simplest procedures, e.g., levee mowing, leads to tall grass and weeds overgrowing. Such conditions favour colonization by burrowing animals, which cause great damage to the levee body. On one hand, tall trees growing near a levee may protect its slopes and top against ice float drafting, but on the other, their deep root system damages the levee and its bed and facilitates water filtration.

The conditions of water seepage through the levee body and subsoil, and the associated phenomena, greatly influence its safe operation. In the event of maximum river water flow rates and levels, usually, within the first few days, water passage into the levee structure is based on filtration. After this period, if the structure still experiences high water levels, seepage occurs primarily through the subsoil, and also commences through the levee body. Water movement in the subsoil ground and levee body can cause the formation of water exudation and effusion zones onto the landside slope and the landside, leading to harmful seepage deformation phenomena [32]. A cross-case analysis based on the course of physical phenomena leading to seepage deformations, as well as their descriptions and definitions was thoroughly discussed in the work [33]. Initial seepage deformations, such as erosion or suffosion can lead to the formation of the most dangerous phenomenon, namely, hydraulic piping. This is why observing the initiation of seepage deformation phenomena is crucial in terms of evaluating levee condition.

Turfing is aimed at protecting the levee against the erosion of its surface. Turf damage caused by animals or ice float drafting can initiate levee slope slides and falls. The sown grasses should exhibit a dense root system, and protect soil particles against washing out from levee slopes.

Waterside slope washouts and slides usually occur in places, where the levee is close to the specific riverbed. Levee waterside slope damage are mainly caused by water waves and ice float drafting. The greatest damage occurs in river meanders, where the levee experiences highest ice float pressure.

**Hydrological factor** characterizes riverbed flow, specifying the flows with a certain probability of occurrence and a consumption curve within the analysed cross-section. The duration of specific flows, as well as riverbed and inter-levee area filling may change over the years. This results from the changes taking place in catchments, such as tightening of the subgrade causing a change in the infiltration coefficient or changes in land development (e.g., deforestation, increased areas used for agricultural purposes or urban development).

Table 1 lists the designations adopted in further analyses for identified and aforementioned key factors influencing the technical condition and safety of levees.

## Results

In order to develop the direct influence matrix *A*, purposive selection of experts was used, through which people competent in the analysis of factors impacting the condition and safety

**Table 1. Analysed factors.**

| Factors | Description |
|---------|-------------|
| F1 | Levee geometry (width, slope inclination, levee top elevation) |
| F2 | Type and condition of soils in levee body |
| F3 | Type, layout and condition of soils in levee bed |
| F4 | Condition of levee areas (inter-levee and landside) |
| F5 | Condition of levee structures |
| F6 | Animal activity |
| F7 | Condition of levee top and slopes (maintenance work, filtration deformations, settlement) |
| F8 | Hydrological factor |

of levees were selected. In order to select competent experts with various knowledge and skills, the following professionals were considered: representatives of scientific centres who are recognised authorities in the hydro-engineering domain, dealing with research and teaching in the field of damming structures, including levees, actively publishing papers and taking part in specialized scientific and technical conferences; representatives of state services assessing the technical condition of levees (in accordance with the Water Law act) and of contractors, who deal with the construction of hydrotechnical facilities, having many years of experience, hold qualifications and references—relevant documents certifying the completion of levee constructions and repairs. In our survey, each expert (two scientists, two representatives of state services assessing the technical condition of levees and two practitioners) conducted pairwise comparisons in terms of influence between factors (Table 1) in the scale 0–3 and provided independently individual direct-influence matrix. In the second stage of the survey, a summary of the responses was presented to the experts and the final matrix was agreed upon during the meeting discussion. Table 2 shows the resulting direct-influence matrix $A$ developed by the experts.

The normalized direct-influence matrix $X$ (Table 3) was calculated from matrix $A$ as a result of applying Eqs (1) and (2). The total-influence matrix $T$ was obtained using Eq (3) and the results are presented in Table 4.

Table 5 lists values of indicators $r_i$ and $c_j$ for analysed factors, calculated using Eqs (4) and (5) and organized in decreasing order. The factor with highest direct and indirect influence is F8 –hydrological factor. The factor with the lowest influence on other factors is F1 –levee geometry. The following factors also exhibit values above the mean in this respect (from highest to lowest): F3 –type, layout and condition soil in levee bed, F4 –condition of levee areas and F7 –condition of levee top and slopes. F2 –type and condition of soils in levee body, F5 –

**Table 2. Direct-influence matrix A.**

| Factor | F1 | F2 | F3 | F4 | F5 | F6 | F7 | F8 |
|--------|----|----|----|----|----|----|----|----|
| F1 | 0 | 1 | 2 | 1 | 0 | 1 | 2 | 0 |
| F2 | 2 | 0 | 1 | 1 | 2 | 1 | 3 | 0 |
| F3 | 2 | 2 | 0 | 3 | 3 | 1 | 2 | 0 |
| F4 | 1 | 1 | 1 | 0 | 2 | 2 | 2 | 2 |
| F5 | 2 | 2 | 1 | 2 | 0 | 1 | 2 | 0 |
| F6 | 2 | 2 | 1 | 1 | 1 | 0 | 2 | 0 |
| F7 | 3 | 3 | 1 | 1 | 2 | 2 | 0 | 0 |
| F8 | 3 | 2 | 2 | 2 | 2 | 1 | 3 | 0 |

**Table 3. Normalized—Direct—Influence matrix (X).**

| Factor | F1 | F2 | F3 | F4 | F5 | F6 | F7 | F8 |
|--------|------|------|------|------|------|------|------|------|
| F1 | 0 | 0.063 | 0.125 | 0.063 | 0 | 0.063 | 0.125 | 0 |
| F2 | 0.125 | 0 | 0.063 | 0.063 | 0.125 | 0.063 | 0.188 | 0 |
| F3 | 0.125 | 0.125 | 0 | 0.188 | 0.188 | 0.063 | 0.125 | 0 |
| F4 | 0.063 | 0.063 | 0.063 | 0 | 0.125 | 0.125 | 0.125 | 0.125 |
| F5 | 0.125 | 0.125 | 0.063 | 0.125 | 0 | 0.063 | 0.125 | 0 |
| F6 | 0.125 | 0.125 | 0.063 | 0.063 | 0.063 | 0 | 0.125 | 0 |
| F7 | 0.188 | 0.188 | 0.063 | 0.063 | 0.125 | 0.125 | 0 | 0 |
| F8 | 0.188 | 0.125 | 0.125 | 0.125 | 0.125 | 0.063 | 0.188 | 0 |

condition of levee structures, and F6—animal activity impact other elements to a degree below the mean. The factor that depends on others (indicator) to the largest degree is F7– condition of levee top and slope, and the one most independent is F8—hydrological factor. Values above the mean are adopted by: F1—levee geometry, F2—type and condition of soils in levee body, F5—condition of levee structures. Values below the mean characterize: F4 –condition of levee areas, F6—animal activity, F3—type and condition of soils in levee body. The position of indicators $r_i$ and $c_j$ for individual factors are shown in the form of an influential power–dependency diagram [34] (Fig 2) (the vertical and horizontal lines described mean values for $C$ and $R$).

Table 5 lists the values of *Prominence* and *Relation* calculated for the analysed factors, in decreasing order. *Relation* enables ranking analysed factors, while the values of *Prominence* describe the role of a factor within a system. Based on Table 5 and Fig 3, it can be seen that the highest value of $r_i+c_i$ is achieved by F7 –condition of levee top and slope, which evidences its key role in the process of determining the total factor influence. F3 –type and condition of soils in levee bed, F4 –condition of levee areas, F5 –condition of levee structure, and F1- levee geometry play an average role in this regard. Values above the mean characterize F2—type and conditions of soils in levee body, while F6—animal activity, reaches a value below the mean. F8—hydrological factor is the least important in this respect.

The interconnection degree for individual factors is quite well demonstrated, if we analyse the reduced form of the total–influence matrix, shown in Table 6. Matrix $T$' was calculated as per Eq (7), where the total influence threshold θ was estimated as the mean for all elements in matrix $T$ and amounts to 0.24. Based on the table, all factors have a significant relationship (>0.24) with F7—condition of levee top and slope, which exhibits the highest *Prominence*.

**Table 4. Total—Influence matrix (T).**

| Factor | F1 | F2 | F3 | F4 | F5 | F6 | F7 | F8 |
|--------|-------|-------|-------|-------|-------|-------|-------|-------|
| F1 | 0.153 | 0.199 | 0.205 | 0.170 | 0.131 | 0.163 | 0.269 | 0.021 |
| F2 | 0.318 | 0.188 | 0.184 | 0.202 | 0.271 | 0.197 | 0.374 | 0.025 |
| F3 | 0.367 | 0.344 | 0.158 | 0.354 | 0.373 | 0.236 | 0.384 | 0.044 |
| F4 | 0.300 | 0.275 | 0.205 | 0.170 | 0.299 | 0.268 | 0.360 | 0.146 |
| F5 | 0.312 | 0.293 | 0.183 | 0.254 | 0.160 | 0.197 | 0.325 | 0.032 |
| F6 | 0.295 | 0.277 | 0.171 | 0.186 | 0.202 | 0.122 | 0.304 | 0.023 |
| F7 | 0.394 | 0.371 | 0.203 | 0.221 | 0.288 | 0.264 | 0.246 | 0.028 |
| F8 | 0.471 | 0.387 | 0.304 | 0.333 | 0.354 | 0.263 | 0.484 | 0.042 |

**Table 5. Total effects and net effects for factor.**

| Order | R | Order | C | Order | R+C | Order | R-C | Group |
|---|---|---|---|---|---|---|---|---|
| F8 | 2.638 | F7 | 2.746 | F7 | 4.761 | F8 | 2.277 | Cause |
| F3 | 2.260 | F1 | 2.610 | F2 | 4.093 | F3 | 0.647 | Cause |
| F4 | 2.023 | F2 | 2.334 | F1 | 3.921 | F4 | 0.133 | Cause |
| F7 | 2.015 | F5 | 2.078 | F4 | 3.913 | F6 | -0.130 | Affected |
| F2 | 1.759 | F4 | 1.890 | F3 | 3.873 | F5 | -0.322 | Affected |
| F5 | 1.756 | F6 | 1.710 | F5 | 3.834 | F2 | -0.575 | Affected |
| F6 | 1.580 | F3 | 1.613 | F6 | 3.290 | F7 | -0.731 | Affected |
| F1 | 1.311 | F8 | 0.361 | F8 | 2.999 | F1 | -1.299 | Affected |

($R_{mean}$ = 1.918; $C_{mean}$ = 1.918; $R+C_{mean}$ = 3.836; grey colour indicates values above the mean value).

The significance of individual factors in respect of *Relation* is shown (Fig 4). The cause group ($r_i$-$c_i$ > 0) includes three factors, namely, F8 –hydrological factor, F3 –type and condition of soils in levee bed, and F4—condition of levee areas. By far, the highest *Relation* value is adopted by F8, which proves that it is the factor that influences other the most. F3 takes second place. The third factor, F4, is characterized by the lowest impact on other factors.

Other factors are F6—animal activity, F5—condition of levee structures, F2—type and condition of soils in levee body, F7—condition of levee top and slopes, and F1—levee geometry belong to the effect group ($r_i$-$c_i$ < 0) and are influenced by causal factors. The factor in this group that is most influenced is F1, F5, F2 and F7 are factors under lesser influence from other factors. F6 is the nearest to the centre, which means that it is least influenced by identified causal factors.

An influential relations map (IRM) developed on the basis of R+C and R-C enables classifying the role of factors within a system (Fig 5). IRM is divided into four squares from I to IV by calculating the mean of (R + C) and 0 for (R-C) [22–24]. Factors in square *I* are identified as core factors or intertwined givers, since they have a high rank and relation; factors in square *II* are classified as driving factor or autonomous drivers, since they have a low rank but high

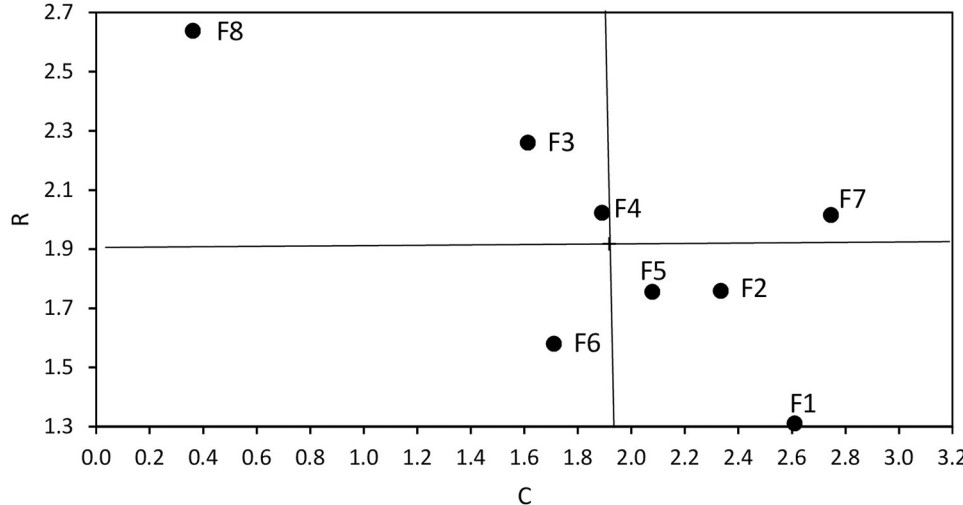

**Fig 2. Diagram of influential power—Dependency.**

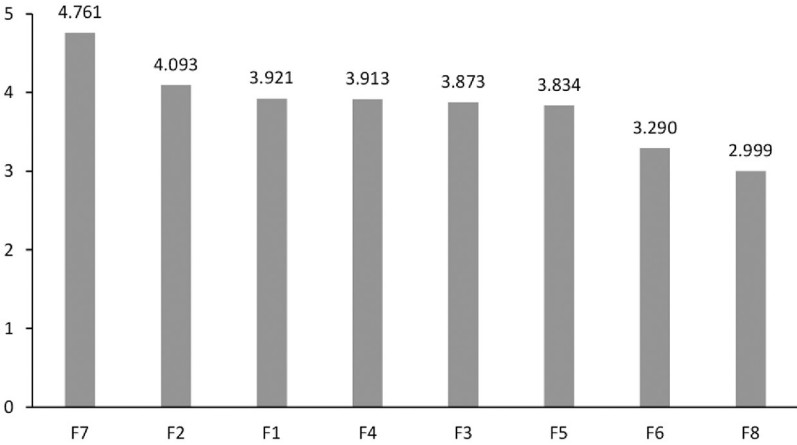

**Fig 3. Prominence graph (R+C).**

relation. Factors in square *III* have a low rank and relation, and are defined as independent factors or autonomous receivers. Factors in square *IV* are characterized by high rank and low relation, and are called impact factors or intertwined receivers.

After applying this classification to the factors in question, it can be concluded that F3 – type and condition of soils in levee body, and F4 –condition of levee areas belong to group *I* (the most important factors), because they impact other factors (are the causes), while simultaneously being significantly related to them ($r_i$-$c_i > 0$; $r_i$+$c_i > mean (R+C))$. F8—hydrological factor belongs to group *II* (important factors), and impact other factors the most, but is poorly related to them. Therefore, it can be treated within a system as an autonomous cause. Group *III* (independent factors) contains two factors, namely, F5 –condition of levee structures and F6—animal activity. They are the effects ($r_i$-$c_i < 0$), and at the same time are poorly connected with other factors. The three remaining factors, F1 –levee geometry, F2—type and condition of soils in levee body, and F7– condition of levee top and slopes belong to group *IV* (indirect factors). They are the effects, while being strongly related to other factors impacting the levee technical condition.

Matrix *T'* (Table 6) was used to also plot relations between factors on the IRM diagram (only values greater than the threshold value are plotted). A solid line is used to represent a significant uni-directional relationship, while a dotted line shows a significant bi-directional relationship. The size of the symbol corresponds to the R+C scale of variation. F7—condition of levee top and slopes has a significant bi-directional relationship with F1—levee geometry, F2—

**Table 6. Matrix *T'*—Reduced form of the total—Influence matrix at a threshold value θ.**

| Factor | F1 | F2 | F3 | F4 | F5 | F6 | F7 | F8 |
|---|---|---|---|---|---|---|---|---|
| F1 | | | | | | | 0.269 | |
| F2 | 0.318 | | | | 0.271 | | 0.374 | |
| F3 | 0.367 | 0.344 | | 0.354 | 0.373 | | 0.384 | |
| F4 | 0.300 | 0.275 | | | 0.299 | 0.268 | 0.360 | |
| F5 | 0.312 | 0.293 | | 0.254 | | | 0.325 | |
| F6 | 0.295 | 0.277 | | | | | 0.304 | |
| F7 | 0.394 | 0.371 | | | 0.288 | 0.264 | 0.246 | |
| F8 | 0.471 | 0.387 | 0.304 | 0.333 | 0.354 | 0.263 | 0.484 | |

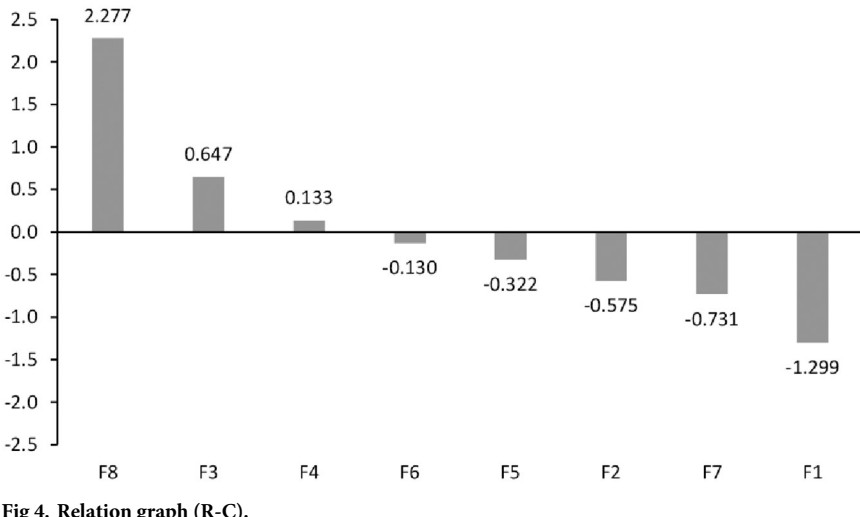

**Fig 4. Relation graph (R-C).**

type and condition of soils in levee body, F5—condition of levee structures, and F6—animal activity. Factors: F8—hydrological factor, F3 –type and condition of soils in levee body, and F4 –condition of levee areas significantly affect its uni-directional influence.

The net influence matrix–*NeT* (Table 7), the elements of which describe the net influence of a given factor and were calculated through Eq (6) was used to assess the influence value of one factor on another. The *NeT* matrix was used to develop an acyclic and asymmetric directed graph called the total net influence map (Fig 6). The developed total factor net influence map definitely confirms the causal nature of F8—hydrological factor, F3—type and condition of soils in levee bed, and F4—condition of levee areas, and the effective nature of F6—animal activity, F5—condition of levee structures, F2—type and condition of soils in levee body, F7—condition of levee top and slopes, and F1—levee geometry. Despite the fact that F3 and F4

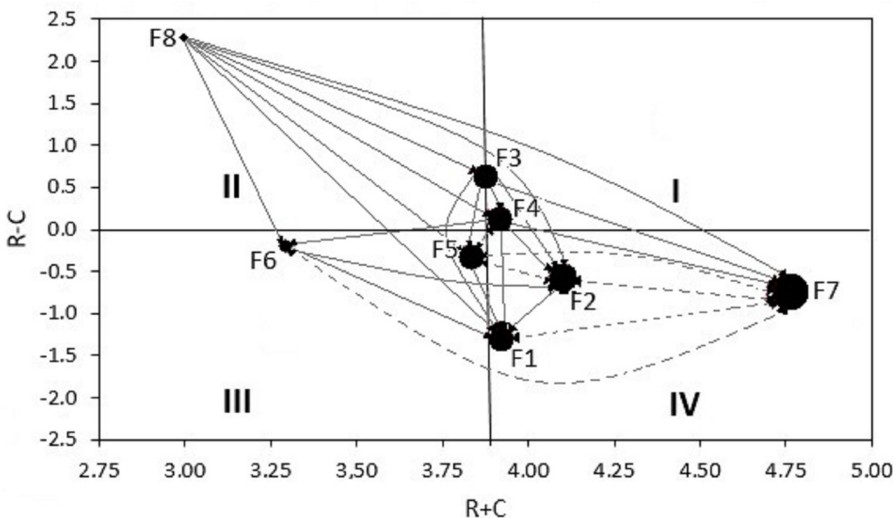

**Fig 5. Influential relation map (IRM) of the factors (mean of R+C = 3.836).**

**Table 7. Net influence matrix (NeT).**

| Factors | F1 | F2 | F3 | F4 | F5 | F6 | F7 | F8 |
|---|---|---|---|---|---|---|---|---|
| F1 | | | | | | | | |
| F2 | 0.12 | | | | | | 0.01 | |
| F3 | 0.16 | 0.16 | | 0.15 | 0.19 | 0.07 | 0.18 | |
| F4 | 0.13 | 0.07 | | | 0.05 | 0.08 | 0.14 | |
| F5 | 0.18 | 0.02 | | | | | 0.04 | |
| F6 | 0.13 | 0.08 | | | 0.01 | | 0.04 | |
| F7 | 0.13 | | | | | | | |
| F8 | 0.45 | 0.36 | 0.26 | 0.19 | 0.32 | 0.24 | 0.46 | |

belong to the cause group, they are impacted by factor F8, which has the highest influence on F3. The F4 factor is also under a slight influence of F3.

## Discussion

Levee failure can result from different type of break mechanisms: internal erosion, overflow, downstream slide, upstream slide and scouring [35]. Each mechanism consists in a sequence of component failures leading to the levee break and is related to the state of eight factors whose model of cause-and-effect relationships we have developed. Generally levee safety assessment is a process at which we evaluate the probability of levee failure, and methods for assessing levee safety are widely discussed in the literature. A wide range of methods are used in assessing levee safety, from the relatively simple based on expert assessment, through index-based methods and empirical models for levee failure modes, to the very complex physical and mathematical models of levee failure modes [36]. A common approach is where the engineer/ expert evaluates qualitatively the performance of levees based on heuristics (knowledge, intuition and experience) [37]. In many cases, experts make levee safety evaluation through

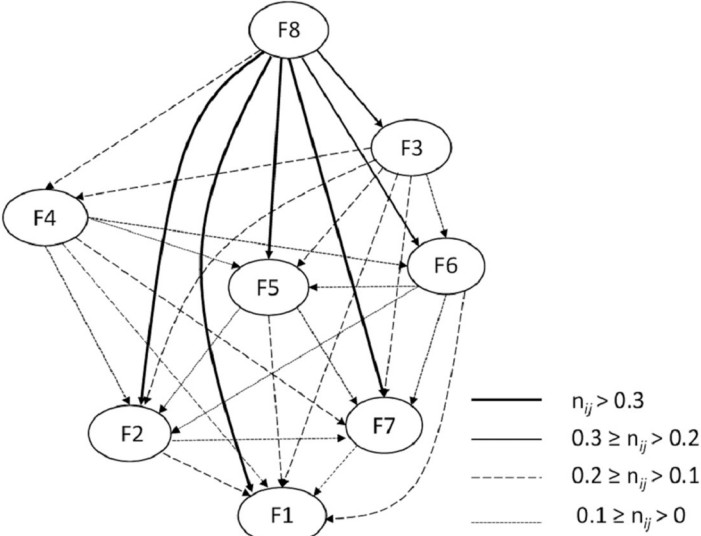

**Fig 6. Map of the total net influence.** Graph arcs indicate a factor under the influence of another factor and the type of line—total influence intensity—the value range for element $n_{ij}$ in the NeT matrix.

probabilistic risk assessment of failure of these structures through direct expression of the numeric data. Fauchard [38] described the principles of a method developed on levees in France that can be successfully applied to levees in other countries around the world. The method usually starts by collecting as much information as possible about the levee based on archival analyses, interviews with managers and a site visit. This phase is essential to ensure the high quality of the final diagnosis. The second stage—geophysical surveys—aims to identify areas of the levee which, due to their specific physical characteristics, are vulnerable to damage during floods. The geophysical methods used must be able to survey over long distances and reveal inhomogeneity both within the levee and its foundation. The third phase, geotechnical investigations—soundings and borings—allow the mechanical characteristics and properties of the materials used to construct the levee to be determined. This approach is representative of those responsible for the safety of flood embankments, in particular, for owners, managers, engineers and contractors. With this in mind, Tourment [39] proposed an analysis of structural failure modes that can help to identify different scenarios of levee failure—and properly evaluate the safety status. The proposed three different assessment methods are based on combining data, expert judgments, index based methods and empirical (physical and mathematical) models. Peyras [40], in contrast, develop methods that can assist levee safety using geographic information system (GIS). Levee safety is closely linked to levee technical condition, which is evaluated using numerous publications in the form of handbooks or guidelines (e.g. [7, 10, 35]), with the most famous being The International Levee Handbook (ILH) [2]. The assessment is based on expert knowledge supported by the results of measurements, analyses, calculations and modelling. The guidelines, depending on the significance class of a hydrotechnical structure, thoroughly define the test, measurement and evaluation procedure, which covers surveying, geotechnical measurement, laboratory analyses and mathematical modelling. The guidelines also contain tips regarding the types and scopes of modelling calculations, starting with calculating slope stability coefficients, and ending with seepage and suffosion potential calculations. Based on the conducted analysis, a traditional levee technical condition evaluation involves the qualitative or quantitative determination of parameters, which characterize the factors (F1-F8) we analysed. Although their inter-relationships are analysed in the performance of levees in the form of sequences leading to levee failure, they are not take into account in the form of the causal-and-effect model that we have developed for these factors. This model cannot, therefore, be regarded as a method to assess levee safety or technical condition. The results we have obtained, allowing through the application of the DEMATEL method, the identifying cause-and-effect relationships between those factors, depend on the form of the direct-influence matrix $A$, that has been formulate by experts, using an integer scale (0–3). A qualitative assessment based on the knowledge, intuition and experience of experts is a common approach in the performance of levees. For example, in the levee safety assessment method developed by Vuillet [37], expert/engineer evaluates analysed criterion of levee performance according to the evaluation scale of 0 to 10. Expert evaluation is also widely used in applications of the DEMATEL method. In an article by Si [17], in which authors reviewed a total of 346 papers on the application of DEMATEL published from 2006 to 2016 in the international journals, only in one case was the direct-influence matrix $A$, developed not on the basis of expert opinion, but as a result of applying the relations map (RM) obtained from the artificial neural network (ANN) [41].

The obtained results from our the causal-and-effect model demonstrate that the factors mostly influencing the technical condition and safety of levees are F8 –hydrological factor, F3 –type and condition of soils in levee body, and F4 –condition of levee areas, which belong to the cause factor group in the case of the developed cause-and-effect model. The hydrological factor is least correlated with other factors, which makes it an autonomous factor with the

highest impact on levee technical condition. Other factors do not influence hydrological phenomena, directly or indirectly, which proves that they cannot impact this factor. Hence, the hydrological factor should be properly estimated at the engineering stage of new structures, and its variability should be closely monitored in relation to old levees. It should be stressed that updating reliable and controlled flow values, and relating them to the values used at the designing stage is becoming more and more desirable. The other two factors, F3 and F4, which have been identified as core factors or intertwined giver are significantly correlated with other factors and are considered as major foundational cause elements. They should be thoroughly assessed at the levee engineering stage and closely monitored, repaired or modernized in the course of facility operation. Their condition determines the state of five other identified factors. The most important phenomena and mechanisms destroying individual levee elements, which result from the type and condition of soils in its bed (F3) include loss of general stability, excessive filtration and associated internal and external erosion [32, 33] that can lead to hydraulic perforation, excessive levee body deformations and settlement, damage to seals extending the filtration route, and possible soil liquefaction [12, 42, 43]. The F3 and F4 factors should be the priorities for engineers, levee managers and decision-makers at the stage of making appropriate decisions in terms of managing the safety of levees as well as their maintenance operations. Their role should be much more emphasized in educational and information campaigns covering flood protection, so that the society is aware that improper utilization of these two elements significantly increases levee unreliability.

Other factors sorted in decreasing order—animal activity (F6), condition of levee structures (F5), type and condition of soils in levee body (F2), condition of levee top and slopes (F7), and levee geometry (F1)—belong to the effect group and are influenced by causal factors. Similar conclusions in terms of relationships between these factors can be found in the work by Hughes [31]. In this group of factors, the levee geometry (F1) is the least influencing factor among all identified ones. The condition of levee top and slopes (F7) has the highest correlation with other factors, which proves its central role in correlations with other factors. F6—animal activity, is least impacted by cause factors. A levee is a hydrotechnical structure, artificially built for protective purposes, and cannot constitute a nature reserve or natural habitat for various, even protected, animal species, but often becomes one due to negligence. Even just mowing grass on a levee leads to reducing the adverse animal activity (F6). A number of other measures are introduced in addition to that. These include incorporating steel grids into the levee body, mainly in order to protect it against beavers. Animal activity (F6) and condition of levee structure (F5) are effect factors that are less related with other factors, and can be treated as autonomous receivers–independent factors. Improving them is a more permanent way to improve the technical condition of a structure. Condition of levee top and slopes (F7), type and conditions of soils in levee body (F2), and levee geometry (F1) are three intertwined receivers in the group of effect factors. They are significantly correlated with others, but their condition depends on causal factors. They should be properly monitored and maintained, but their modernization or repair, without a simultaneous elimination of irregularities in causal factors, do not lead to effective, long-term improvement of levee technical condition and safety. The group of phenomena and factors that depend on the type and conditions of soils in the levee body (F2) include loss of general or local stability, excessive filtration, internal and external erosion, hydraulic perforation, structure-substrate interface damage, soil liquefaction, and the destructive activity of animals and plants [13, 42]. Deformation of the levee body and bed are most usually caused by a defective structure and the application of unsuitable soil. This mainly applies to levees executed using old technologies, when using soils found in near proximity from the route of the constructed levee was often the case. In such cases, the levee

structure is characterized by heterogeneity of incorporated soils, and their insufficient or uneven compaction.

## Conclusions

Our analyses allow us to formulate the following conclusions:

1. The discussed in-depth cause-and-effect analysis of factors influencing levee technical condition and safety, conducted using the DEMATEL technique, enabled classifying them as cause or effect factors.

2. The cause factor group includes, in decreasing order, the hydrological factor (F8), type and condition of soils in levee bed (F3) and condition of levee areas (F4). The hydrological factor, despite being a primary causal factor, is least correlated with other factors, which makes it an autonomous factor with the highest impact on levee technical condition. Other factors do not influence hydrological phenomena, directly or indirectly, which proves that they cannot impact this factor.

3. Other factors sorted in decreasing order—animal activity (F6), condition of levee structures (F5), type and condition of soils in levee body (F2), condition of levee top and slopes (F7), and levee geometry (F1)—belong to the effect group and are influenced by causal factors. In this group of factors, the levee geometry (F1) is the least influencing factor among all identified ones. The condition of levee top and slopes (F7) has the highest correlation with other factors, which proves its central role in correlations with other factors. Condition of levee top and slopes (F7), type and conditions of soils in levee body (F2), and levee geometry (F1) are three intertwined receivers in the group of effect factors. They are significantly correlated with others, but their condition depends on causal factors. They should be properly monitored and maintained, but their modernization or repair, without a simultaneous elimination of irregularities in causal factors, do not lead to effective, long-term improvement of levee technical condition and safety.

4. The results obtained through the DEMATEL method are consistent with engineering intuition and experience. However, the calculations demonstrate interdependencies and determine their significance in a quantitative manner, which is not always possible in practice.

## Supporting information

**S1 File.**
(XLSX)

## Author Contributions

**Conceptualization:** Dorota Mirosław-Świątek, Paweł Popielski, Piotr Śliwiński, Zdzisław Skutnik.

**Data curation:** Tomasz Cwalina.

**Formal analysis:** Dorota Mirosław-Świątek, Paweł Popielski.

**Methodology:** Dorota Mirosław-Świątek, Paweł Popielski, Piotr Śliwiński, Zdzisław Skutnik.

**Validation:** Dorota Mirosław-Świątek.

**Visualization:** Dorota Mirosław-Świątek, Tomasz Cwalina.

**Writing – original draft:** Dorota Mirosław-Świątek, Paweł Popielski, Piotr Śliwiński, Zdzisław Skutnik.

**Writing – review & editing:** Dorota Mirosław-Świątek, Paweł Popielski, Piotr Śliwiński, Zdzisław Skutnik.

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
