## [Decision Letter · Decision Letter 0]

28 May 2021

PONE-D-21-09108

Analysis of factors influencing levee safety using the DEMATEL method

PLOS ONE

Dear Dr. Mirosław-Świątek,

Thank you for submitting your manuscript to PLOS ONE. After careful consideration, we feel that it has merit but does not fully meet PLOS ONE’s publication criteria as it currently stands. Therefore, we invite you to submit a revised version of the manuscript that addresses the points raised during the review process.

Please consider all the comments of all reviewers

We look forward to receiving your revised manuscript.

Kind regards,

Ahmed Mancy Mosa, Ph.D.

Academic Editor

PLOS ONE

Journal Requirements:

Reviewers' comments:

Reviewer's Responses to Questions

**Comments to the Author**

1. Is the manuscript technically sound, and do the data support the conclusions?

Reviewer #1: Yes

Reviewer #2: Yes

2. Has the statistical analysis been performed appropriately and rigorously? 

Reviewer #1: N/A

Reviewer #2: Yes

3. Have the authors made all data underlying the findings in their manuscript fully available?

Reviewer #1: Yes

Reviewer #2: Yes

4. Is the manuscript presented in an intelligible fashion and written in standard English?

Reviewer #1: Yes

Reviewer #2: Yes

5. Review Comments to the Author

Reviewer #1: The article entitled “Analysis of factors influencing levee safety using the DEMATEL method” aims to apply the decision-making trial and evaluation laboratory (DEMATEL) method to develop a cause-and-effect model for factors impacting the condition and safety of levees.

The paper is well written and properly organized. The English is correct.

The authors introduce the paper correctly and explain the study. They use a methodology, which is not novel, and apply it to study the safety level of levees. The safety is assessed by means of experts’ knowledge. Therefore, the authors do not quantify this safety level. This is the main drawback of the paper. In order to prove the results obtained in the study, they should be related to some other studies performed on the assessment of levees conditions.

The manuscript could improve its scientific base and novelty by including information regarding other studies but also, by including the results of those studies in the elaboration of matrices, if possible.

There is no mention about the piping effect, which is important in the safety assessment of levees.

My recommendation is major revision.

Reviewer #2: The authors did a good work and covered an interesting area. However, only few remarks can be addressed to improve the quality of the article:

Conclusions: Rewrite the conclusions in Points

Double check the references and unify the citation style in the list

6. PLOS authors have the option to publish the peer review history of their article (what does this mean?). If published, this will include your full peer review and any attached files.

Reviewer #1: No

Reviewer #2: No

---

## [Author Response · Author response to Decision Letter 0]

14 Jul 2021

Reviewer 1

We would like to thank the Reviewer for the work done in this revision. We will follow and answer all the proposed questions/issues to correctly improve the manuscript.

They use a methodology, which is not novel, and apply it to study the safety level of levees. The safety is assessed by means of experts’ knowledge. Therefore, the authors do not quantify this safety level. This is the main drawback of the paper. In order to prove the results obtained in the study, they should be related to some other studies performed on the assessment of levees conditions. The manuscript could improve its scientific base and novelty by including information regarding other studies but also, by including the results of those studies in the elaboration of matrices, if possible.

Response:

We would like to note kindly that the aim of our study was to find out the inter-relationships of factors impacting the levee technical condition performed as a cause-and-effect model and not to develop a method for assessing levee safety of levees conditions. The obtained results from our the causal-and-effect model allowing through the application of the DEMATEL method, the identifying cause-and-effect relationships between those factors and classifying them as cause or effect factors. The results of this type of analysis have not been presented in the literature so far. This approach cannot be interpreted as a method for evaluating the performance of levees. Therefore, our results cannot be related to some other studies performed on the assessment of levees conditions. In revised manuscript the Discussion has been rewritten as suggested by the Reviewer. We clarified better the context of our analyses in relation to levee performance evaluation methods (lines 374-424). 

In the revised manuscript the following articles have been added to the references:

1. Mériaux P, Royet P, Folton C. 2001. Surveillance, entretien et diagnostic des digues de protection contre les inondations [Monitoring, maintenance and diagnosis of dikes: A practical guide for owners and managers]. Paris: Cemagref ed.

2. FloodProbe. 2012. CombiningInformationforUrbanLevee Assessment, ReportNumber:WP03-01-12-24.Availablefrom: http://www.floodprobe.eu

3. Vuillet M, Peyras L, Carvajal C, Serre D, Diab Y. 2013. Levee performance evaluation based on subjective probabilities, European Journal of Environmental and Civil Engineering, 17:5, 329-349, DOI: 10.1080/19648189.2013.785723

4. Fauchard C, Mériaux P. 2007. Geophysical and Geotechnical Methods for Diagnosing Flood Protection Dikes, Guide for implementation and interpretation.

5. Tourment R., Beullac B., Sab G A. 2019. Levee safety assessments using an index-based method. XVII ECSMGE-2019, Geotechnical Engineering foundation of the future, Sep 2019, Reykjavik, Iceland. pp.8.

6. Peyras L, Tourment R, Vuillet M, Beullac B, Delaunay C., Bambara G. 2017. Development of an expert-led GIS-based approach for assessing the performance of river levees: the D igsure method and tool. Journal of Flood Risk Management 10:3, pages 393-407.

7. Chen Y, Liu J, Li Y, Sadiq R, Deng Y. 2015.RM-DEMATEL: a new methodology to identify the key factors in P M 2.5, Environmental Science and Pollution Research, vol. 22, no. 8, pp. 6372–6380.

There is no mention about the piping effect, which is important in the safety assessment of levees.

Response:

Hydraulic piping is one of the most most dangerous phenomenon, which leads to a levee failure. It was not included as a factor influencing levee technical condition because its mechanism of occurrence is related to a sequence of events resulting from the condition of factors F7 - condition of levee top and slopes (maintenance work, filtration deformations, settlement) and F8 - hydrological factor, which were included in our analysis. Its mechanism of formation in relation to the F7 and F8 factor is explained in revised manuscript (lines 227-238). 

Reviewer 2

We would like to thank the Reviewer for the work done in this revision. We will follow and answer all the proposed questions/issues to correctly improve the manuscript.

The authors did a good work and covered an interesting area. However, only few remarks can be addressed to improve the quality of the article: Conclusions: Rewrite the conclusions in Points. Double check the references and unify the citation style in the list.

Response:

As suggested by the Reviewer we have rewritten the Conclusions in points. In the revised manuscript we have corrected the citation style the reference.

---

## [Decision Letter · Decision Letter 1]

23 Jul 2021

Analysis of factors influencing levee safety using the DEMATEL method

PONE-D-21-09108R1

Dear Dr. Mirosław-Świątek,

We’re pleased to inform you that your manuscript has been judged scientifically suitable for publication and will be formally accepted for publication once it meets all outstanding technical requirements.

Kind regards,

Ahmed Mancy Mosa, Ph.D.

Academic Editor

PLOS ONE

Additional Editor Comments (optional):

Reviewers' comments:

Reviewer's Responses to Questions

**Comments to the Author**

1. If the authors have adequately addressed your comments raised in a previous round of review and you feel that this manuscript is now acceptable for publication, you may indicate that here to bypass the “Comments to the Author” section, enter your conflict of interest statement in the “Confidential to Editor” section, and submit your "Accept" recommendation.

Reviewer #1: All comments have been addressed

2. Is the manuscript technically sound, and do the data support the conclusions?

Reviewer #1: Yes

3. Has the statistical analysis been performed appropriately and rigorously? 

Reviewer #1: Yes

4. Have the authors made all data underlying the findings in their manuscript fully available?

Reviewer #1: Yes

5. Is the manuscript presented in an intelligible fashion and written in standard English?

Reviewer #1: Yes

6. Review Comments to the Author

Reviewer #1: The authors have addressed all the comments proposed in the revision. There are no further comments.

7. PLOS authors have the option to publish the peer review history of their article (what does this mean?). If published, this will include your full peer review and any attached files.

Reviewer #1: No

---

## [Editor Report · Acceptance letter]

17 Aug 2021

PONE-D-21-09108R1 

Analysis of factors influencing levee safety using the DEMATEL method 

Dear Dr. Mirosław-Świątek:

I'm pleased to inform you that your manuscript has been deemed suitable for publication in PLOS ONE. Congratulations! Your manuscript is now with our production department. 

Kind regards, 

on behalf of

Dr. Ahmed Mancy Mosa 

Academic Editor

PLOS ONE